# Molecular Mechanism in the Development of Pulmonary Fibrosis in Patients with Sarcoidosis

**DOI:** 10.3390/ijms241310767

**Published:** 2023-06-28

**Authors:** Elisabetta Cocconcelli, Nicol Bernardinello, Gioele Castelli, Simone Petrarulo, Serena Bellani, Marina Saetta, Paolo Spagnolo, Elisabetta Balestro

**Affiliations:** Respiratory Disease Unit, Department of Cardiac, Thoracic, Vascular Sciences and Public Health, Padova University Hospital, 35128 Padova, Italygioele.castelli@studenti.unipd.it (G.C.); serena.bellani@studenti.unipd.it (S.B.); paolo.spagnolo@unipd.it (P.S.)

**Keywords:** sarcoidosis, advanced pulmonary fibrosis, signaling, molecular pathway

## Abstract

Sarcoidosis is a multisystemic disease of unknown etiology characterized by the formation of granulomas in various organs, especially lung and mediastinal hilar lymph nodes. The clinical course and manifestations are unpredictable: spontaneous remission can occur in approximately two thirds of patients; up to 20% of patients have chronic course of the lung disease (called advanced pulmonary sarcoidosis, APS) resulting in progressive loss of lung function, sometimes life-threatening that can lead to respiratory failure and death. The immunopathology mechanism leading from granuloma formation to the fibrosis in APS still remains elusive. Recent studies have provided new insights into the genetic factors and immune components involved in the clinical manifestation of the disease. In this review we aim to summarize the clinical-prognostic characteristics and molecular pathways which are believed to be associated with the development of APS.

## 1. Introduction

Sarcoidosis is a systemic disorder characterized by granulomatous reactions related to an unknown cause and occurring in any organ, likewise lung, eye, liver, nervous system, skin, spleen and also heart, even though the pulmonary involvement is recognized in nearby the totality of affected patients [1].

As many other interstitial lung diseases (ILDs) [2,3,4,5] the clinical course of sarcoidosis is unpredictable with some patients totally asymptomatic and others presenting clinical progression with loss of function of the affected organs. The development of clinically significant disease depends on whether granulomatous inflammation resolves—either spontaneously or with treatment—or persists and progresses to fibrosis and organ failure.

The term advanced pulmonary sarcoidosis (APS) refers to pulmonary fibrosis, complicated with bronchiectasis and/or infections, and pulmonary hypertension. Although cardiac and neurological involvement are common causes for sarcoidosis-related death, APS is the leading cause of poor outcomes and responsible for increased hospitalization and mortality rates among affected patients both in Europe and in US [6,7,8] (https://doi.org/10.1253/jcj.64.679, accessed on 20 June 2023) (https://doi.org/10.3389/fmed.2023.999066, accessed on 20 June 2023). During the pandemic, patients with APS were considered at higher risk of mortality because of the multi systemic inflammatory environment caused both by SARS-CoV-2 infection and sarcoidosis (https://doi.org/10.3390/pathogens11080867, accessed on 20 June 2023).

Numerous genome-wide association studies (GWAS) performed in sporadic and familial cases of sarcoidosis (i.e., cases affecting two or more members of the same biological family) suggested a synergistic role of different immune pathways [9]. Polymorphisms in immune-related genes contribute not only to the pathological granuloma formation, but also to the entire and highly variable course of the disease.

Moreover, many key cells like T cells and macrophages are involved in granuloma formation. Persistent inflammation is suggested to be a risk factor or a precursor of irreversible fibrosis. However, many authors assert that chronic sarcoidosis is not simply the natural consequence of the acute phase for sarcoidosis, because not each patient with chronic inflammation will develop fibrosis. To date, the effective link between granuloma formation and fibrosis in APS remains still unsolved. Corticosteroids are the most common first-line drugs used, while the antimetabolites and biological agents representing an alternative second and third-line choice for patients who failed with corticosteroids or who cannot tolerate them [1,10].

Recently antifibrotic medication (nintedanib) has been approved in progressive fibrotic lung diseases including sarcoidosis, once other immunosuppressive therapies have been exhausted [3]. Nonetheless, the absence of reliable predictors of disease progression and the scarcity of effective drugs greatly contribute to making sarcoidosis a difficult disease to manage and to prevent the evolution into pulmonary fibrosis.

This review focuses on APS addressing its clinical-prognostic characteristics and molecular pathways that are believed to explain this progressive fibrosing lung disorder.

## 2. Clinical Presentation

Pulmonary involvement is recognized in nearby the totality of patients affected by sarcoidosis, even though most patients are asymptomatic or present symptoms suggestive for a systemic involvement. APS with fibrosis occurs in 10–20% of sarcoid patients, and it is the main cause of respiratory failure and death [6].

There are no typical symptoms of fibrotic pulmonary sarcoidosis as dry cough, dyspnoea on effort and chest discomfort are common presenting symptoms for patients with pulmonary sarcoidosis without significant fibrosis [1]. In addition, such symptoms may suggest different diagnoses such as chronic bronchitis or other ILDs, therefore the definite diagnosis of advanced pulmonary fibrosis could be delayed. Differently from other ILDs, only a minority of sarcoid patients presents velcro-like crepitations at chest auscultation or digital clubbing [11].

Pulmonary function tests of advanced sarcoid patients present both restrictive and obstructive patterns, sometimes even with a mixed pattern [1,11]. However, the most common feature is the reduction of diffusing capacity for carbon monoxide (DLCO), which in sarcoidosis worsens with the increase of lung impairment [12].

Pulmonary hypertension (PH) is a typical complication in 5–28% of patients with pulmonary sarcoidosis and occurs especially in up to 74% of patients with advanced sarcoidosis [13]. This complication is particularly associated with exertional dyspnea in nearly 50% of cases and leads to a worst prognosis [14,15]. Multiple conditions are possible causative agents and include hypoxemia due to ILD, vascular disease, mediastinal distortion/compression from lymphadenopathy, and extrapulmonary diseases (e.g., left ventricular dysfunction). This complication leads to a higher possibility of listing for lung transplantation, but also an higher risk of mortality while on the waiting list. The presence of sarcoidosis-associated pulmonary hypertension, is known to worsen outcomes. Indeed it has been reported a 7-fold increased risk for death over 3-year follow-up in one case series [14] and more recently a mortality rate over median 3-year follow-up of 32% [13]. Right heart catheterization is the gold standard in diagnosing PH, but the finding of a main pulmonary artery to descending aorta ratio of >1 may be used as a reliable marker for pulmonary hypertension.

Some clinical parameters have been suggested as prognostic predictors for mortality in patients with APS, in particular the occurrence of respiratory failure has been considered the most common cause of sarcoidosis-related death [1]. Identifying patients with a worse prognosis has several useful implications and guides management decisions. First, new therapies for patients who have failed the conventional anti-inflammatory drugs could be early explored. Second, delays in referring patients for lung transplant, with a shortened pre-transplant and post-transplant survival, could be avoided.

In 2011, Nardi A. and colleagues described the causes of death in 142 advanced pulmonary sarcoid patients compared to a matched general population along a ten years follow up. During the observational period, PH developed in 30% of APS patients and sixteen of 142 patients died. Main causes of deaths were refractory PH and respiratory failure [11].

In 2017, Kirkil G. and colleagues described which clinical variables may be associated with increased mortality in a cohort of 452 sarcoid patients [7]. The overall mortality was 3.9% and 9% at 5 and 10 years of follow up, respectively. In 90% of cases, death was related to respiratory failure, whereas non-sarcoidosis related causes (cancer and coronary artery diseases) were recognized in the other 10% of cases. Age, black race, stage IV at chest X-ray, the presence of 20% or more of fibrosis on high resolution CT scan (HRCT), or PH were associated with increased risk for mortality, with age, extent of fibrosis on HRCT scanning and PH independent predictors for it. More recently a retrospective study of 216 patients with APS and a median follow-up period of 8 years demonstrated that fibrosis involving greater than 20% of the lung parenchyma was associated with a median of 8 years for transplant-free survival compared with 17 years for those with fibrosis involving less than or equal to 20% of the lung parenchyma [16].

## 3. Imaging

Imaging plays a central role in both the diagnosis and the follow up of pulmonary sarcoidosis patients. The advanced pulmonary sarcoidosis is synonymous to stage IV presentation of the Scadding score, with pulmonary fibrosis visible at chest radiography [17]. Recently, HRCT scans had progressively substituted chest X-ray in the definition and diagnosis of sarcoidosis [18]. Fibrosis typically occurs in the upper lobes, radiating from the hilum, associated with architectural distortion, especially in central bronchovascular areas. Specifically, the main HRCT scan features of fibrotic sarcoidosis leading to chronic respiratory failure are consolidations along the bronchovascular bundles comprising central-peripheral bands, traction bronchiectasis and upper lobe shrinkage with usually a posterior displacement of the main or upper-lobe bronchus and volume loss [19]. In a minority of cases honeycombing may be detected, with a radiological appearance similar to a usual interstitial pneumonia (UIP) pattern [20,21].

In patients with sarcoidosis the evaluation of Right Upper Lobe Bronchus Angle (RUL-BA) on chest HRCT may be important to evaluate fibrotic (i.e., stage IV) versus non-fibrotic disease. RUL-BA is a measure of the angle between a line traversing the right upper lobe bronchus and a sagittal line connecting the sternum to the vertebral body and tangential to the most medial aspect of bronchus. RUL-BA may assist both expert radiologists and clinician in detecting APS even if further research is needed to replicate this radiologic tool in a larger population [22].

More recently, in a retrospective cohort study of 106 sarcoidosis patients with APS, Schimmelpennink and co-authors showed that UIP-like pattern on HRCT is an independent predictor for all-cause mortality and lung transplantation [23]. To date, the extent of fibrosis on HRCT is considered a prognostic predictor not only in ILDs as IPF, but also in sarcoid patients [7]. HRCT findings, and more specifically the quantification of fibrosis and honeycombing on the CT scan, have been widely suggested to correlate with disease severity and prognosis [18,21,24,25,26,27].

Fluoro-deoxyglucose-positron emission tomography (FDG-PET) is a useful tool in identifying patients with an active disease and to differentiate them from those with stable fibrotic changes [24].

Mycetomas, masses of fungal mycelia commonly of *Aspergillus* spp., are present in 3–12% of end-stage pulmonary sarcoidosis and are known as chronic pulmonary aspergillosis [28,29]. This complication appears almost exclusively in stage IV sarcoidosis and it is associated with a worse outcome. Mycetomas are visible at CT scans as soft-tissue density masses, with a distribution similar to the fibrotic scars in the upper lobes [20]. Typically asymptomatic, this occurrence can be associated with haemoptysis, ranging from minimal to massive and life threatening [29].

## 4. Genetic Factors

The incidence of sarcoidosis in Northern European countries counts up to 60 cases per 100,000 people, in contrast with a lower incidence count observed in the South European regions. These geographic variations in sarcoidosis incidence strongly indicate the linkage between genetic and environmental factors, including smoking. Of interest, a study by Rivera et al. performed in 747 Swedish sarcoidosis cases suggested that the effect of smoking as a risk factor for sarcoidosis is modulated by carriage of certain genetic variants, and highlights the importance of integrating genetic information when assessing the relationship between sarcoidosis and environmental exposures [30].

Numerous GWAS performed in sporadic and familial cases of sarcoidosis (i.e., cases affecting two or more members of the same biological family) suggested a synergistic role of different immune pathways [9,31]. Polymorphisms in immune-related genes contribute not only to the pathological granuloma formation, but also to the entire and highly variable course of the disease. The specific human leukocyte antigen (HLA) class II haplotypes, innate immunity-related genes, regulators of calcium channels, G-protein-coupled receptors, the mammalian target of rapamycin (mTOR)-related pathways, the ras-related C3 botulinum toxin substrate 1 (Rac1) hubs, *PTPRD*, *FAT* atypical cadherins and *KIF* genes, regulator of migration and survival, are the most strongly associated genetic risk factor for sarcoidosis, supporting the opinion that sarcoidosis is an exposure-mediated immunologic disease [32,33,34]. Some single nucleotide polymorphisms (SNP) has been associated to genetic susceptibility for fibrotic sarcoidosis, such as genes encoding for gremlin, transforming growth factor-beta3, prostaglandin-endoperoxide synthase 2, and caspase recruitment domain15 [35,36,37,38]. Interestingly, the well-known SNP in the promoter of mucin-5B gene seems not to be associated with the fibrotic presentation in sarcoidosis [39]. Among few studies concerning familial relative risks for sarcoidosis [40,41]. Rossides M. and co-authors, by using population-based registers, estimated valid and precise familial aggregation and heritability estimates and showed that familial exposure to sarcoidosis is a very strong risk factor for the disease. The heritability of the disease was 39%, suggesting a stronger implication of environmental factors in the development of sarcoidosis [42]. 

## 5. Main Key Players of Fibrotic Sarcoidosis

Granulomas formation represents a fixed hallmark in sarcoidosis [43]. It is well known that CD4+ T cells and macrophages are the key cells of the inflammatory response in sarcoidosis, occurring in genetically susceptible individuals after the inhalation of unknown antigens, probably of mycobacterial nature [44]. In the acute form of sarcoidosis, the antigens induce the activation of dendritic cells (DCs) and consequently their migration in mediastinal lymph nodes, becoming antigen-presenting cells (APCs). T cells are subsequently recruited by APCs with class I and II histocompatibility complex (MHC) molecules. Once activated by antigens, macrophages differentiate themselves into epithelioid cells, producing a well-organized concentric structure to stem the causal antigen. Finally, a rim of B lymphocytes, dendritic cells, T cells and fibroblasts surround the granuloma and protect the central core [45]. However, the link between granuloma formation and fibrosis in APS remains still unsolved. Recent studies using FDG-PET scans highlighted that, in some patients, inflammation and fibrosis could exist [24]. Persistent inflammation is suggested to be a risk factor or a precursor of irreversible fibrosis. However, many authors assert that chronic sarcoidosis is not simply the natural consequence of the acute phase for sarcoidosis, because not each patient with chronic inflammation will develop fibrosis. During the disease’s natural history, in some patients profibrotic events are mixed with an innate susceptibility. Moreover, several studies demonstrated that T helper (Th)1 lymphocyte immunity changes in favor to Th2 milieu [46] and the macrophage phenotype switches from M1 to M2 [47]. These are only some key features of this complex interplay between adaptive and innate immunity in fibrotic sarcoidosis. In Table 1 are summarized the selected genetic, key cells type and the main signaling pathways which have been reported to be associated with fibrotic sarcoidosis as explained in details afterwards.

## 6. T Cells

During the acute form of sarcoidosis, higher levels of interleukin (IL)-2, interferon (INF)-γ and IL-12 are present in the lung of affected patients, and a huge amount of lymphocytes with an elevated CD4/CD8 ratio and pro-inflammatory cytokines [such as IFN-γ, tumor necrosis factor (TNF)-a and macrophage inflammatory protein-1b (MIP-1b)] are detectable in the bronchoalveolar lavage (BAL) fluid [48].

The polarization of T cells in Th1 type via MHC class II molecules is essential for granuloma formations [49,59]. Th1 cells present an abnormal T cell receptor (TCR) signaling and a sort of oligoclonal dysfunction [50,60]. This mechanism is probably caused by a higher expression of programmed death-1 (PD-1) and other factors that negatively regulate the correct polyclonal proliferation of T cells. The subsequent role of T-cells in developing pulmonary fibrosis remains difficult to understand. Some evidence reports that the M1/Th1 pathway (typical of the acute phase) may change towards a pro-fibrotic M2/Th2 axis, leading to a pro-fibrotic pathway, even though some aspects remain to be elucidated [61,62]. In fact, the presence of increased level of specific Th2 cytokines (such as IL-4, IL-5, IL-9, IL-13, IL-10 and TGF-β) and reduced levels of INF-γ, are associated with extracellular matrix (ECM) production [63]. Among others, IL-13 can improve the production of TGF-β and can suppress TNF-a release. However, in a study conducted by Hauber and co-workers, IL-13 level was higher in patients without pulmonary fibrosis [64] identifying an important area for further investigation. On the contrary, Patterson and co-workers found a higher level of IL-5 and altered levels of IL-7 and GM-CSF in pulmonary fibrosis, supporting the transition hypothesis [65]. Recently, among 465 patients with biopsy-proven sarcoidosis, it has been shown the utility of neutrophil–lymphocyte ratio (NLR) in predicting advanced disease stage and discriminating between active and stable disease [66]. However further studies are needed to understand the role of the Th1/Th2 axis in developing fibrosis in patients with pulmonary sarcoidosis. Other immunological interplay like chemokine ligand 2 (CCL-2) and chemokine ligand 5 (CCL-5) can improve fibroblast survival and are elevated in sarcoid lungs. However, the role of CCL-2 and CCL-5 in the pro-fibrotic pathway was not evaluated in specific phenotyping patients with stage IV sarcoidosis [67]. 

## 7. Macrophages

Phagocytosis is the main role of alveolar macrophages. In the alveolar spaces, macrophages remove microorganisms, debris and old unfunctional cells [68]. Macrophages could be activated in two different forms: the M1 or “killer” form or, alternatively, the M2 or “healer” form. When stimulated by an external antigen, macrophages can initiate a strong inflammatory cascade, promoting the switch from innate to adaptive immune response. In sarcoidosis, the number of macrophages is generally increased in the lung with high production of several mediators, such as TNF-a and are considered to be central in sarcoidosis pathogenesis as they activate T-cells, produce pro-inflammatory cytokines that drive inflammation [69]. Of interest, in a large cohort of sarcoidosis patients, it has been observed that the frequency and distribution of monocytes in blood and BAL at time of diagnosis may predict disease outcome and high frequencies of TNF producing monocytes/monocyte-derived cells are associated with progressive disease development [70]. Recently, it has been observed that in sarcoidosis patients, monocyte subsets have distinct lower expression of regulatory receptors. They present a reduced CD200R and CD47 expression compared with healthy controls, with consequently an increased proliferative rate [71]. Moreover, in sarcoidosis with pulmonary fibrosis a polarization from M1 to M2 has been demonstrated. Th2 cells release IL-4 and IL-13 that stimulate the proliferation of M2, potent producers of TGF-β and other signals that could stimulate fibroblasts’ activity [51,52,72]. Also chemokine ligand 18 (CCL-18) are overexpressed during M2/Th2 recruitment and seems to be specifically associated with pulmonary fibrosis, but not with other forms of pulmonary involvement [73]. Another mechanism that contributes to fibrosis is the arginine pathway. M2 can over produce arginase through the expression of the Arg1 gene. In this way, arginine is converted to ornithine, a precursor of collagen [74]. The M2 polarization is also reported to play a key role in the development of fibrosis in neuromuscular sarcoidosis [75]. In neuromuscular sarcoidosis, authors found an increased expression of CD206, CD301 and Arg-1. Depending on different signals, macrophages are more flexible in comparison with T-cell polarization and their different activation, as needed, can be reversible. However, new studies with specific disease phenotyping are needed for a correct interpretation of the M1/M2 axis.

## 8. Signaling Pathway in Fibrotic Sarcoidosis

### 8.1. TGF-β/Smad Signaling

TGF-β/Smad signaling represents an important key pathway during inflammatory mitigation and wound healing [76]. Moreover, TGF-β/Smad abnormalities are very common, not only in respiratory disorders, but also in other fibrotic extra-pulmonary diseases, showing a promising role in the development of new target therapies [76,77,78,79]. Three different forms of TGF-β were described in mammals and each of them seems to be differently implicated in the mechanism of fibrosis. In particular the first discovered, TGF-β1, is strongly connected with fibrosis and presents a central role in collagen deposition, fibroblast recruitment and myofibroblast transformation [53,80]. On the contrary, the role of TGF-β3 is still unsolved [81] whereas TGF-β2 seems to be implicated in post inflammatory wound healing through EGFR phosphorylation [54]. Recently, some studies elucidate the role of TGF-β/Smad 3 signaling in the progression from acute form of sarcoidosis to chronic fibrosis. For example, Bonniaud P. et al. demonstrated that TGF-β1/Smad2 signaling was strongly positive in fibrotic areas observed in WT mice at 35 days of proinflammatory AdIL-1beta administration. Indeed, a more severe disease and parenchymal involvement seems to be associated with a higher expression of TGF-β/Smad 3 signaling [55,82].

### 8.2. JAK-STAT Signaling

The release of IFN-γ is regulated by the Janus kinase/signaling transducer and activator of transcription (JAK/STAT) pathway [83]. To date, four JAKs (JAK 1, 2, and 3, and TYK2) and seven STATs (STAT 1, 2, 3, 4, 5A, 5B, and 6) have been reported in literature [83]. JAK/STAT pathway is over expressed in patients with sarcoidosis in comparison with healthy subjects. Zhou T. and coworkers revealed that JAK/STAT is the most significantly represented in their 17-gene signature analysis. Moreover, JAK/STAT expression is higher in patients with more complicated sarcoidosis and is also higher in patients with mild sarcoidosis compared with healthy subjects [56]. However in this study the difference between complicated sarcoidosis and fibrotic pulmonary sarcoidosis is not clear. Of interest, a dramatic response with ruxolitinib was described in a patient with multisystemic sarcoidosis and moderate pulmonary fibrosis with JAK2-mutated polycythemia [84].

### 8.3. mTOR Signaling

The protein kinase mTOR (mechanistic target of rapamycin) is involved in many regulatory and proliferation mechanisms, forming two different complexes called mTOR complex 1 and mTOR complex 2 (mTORC1 and mTORC2) [85] mTOR is implicated in several diseases such as tuberous sclerosis (TSC), cancer, diabetics and obesity, among others [86]. Moreover, the activation of the PI3K/AKT/mTOR signaling promotes collagen synthesis in the lung fibroblasts through aerobic glycolysis [57] and represents a powerful autophagy inhibitor. mTOR was recently investigated in sarcoidosis, however with contrasting results. First, PD-1 pathway effects in PI3K/AKT/mTOR signaling. More specifically, the blockade of PD-1 in sarcoidosis induces a restoration of correct T cell proliferation and the normal expression of PI3K/AKT/mTOR signaling [87]. Second, in 2017 Linke and co-workers found that mTORC1 checkpoint induced granuloma formation when the TSC2 inhibitor was depleted [85]. Moreover, they found an association between mTOR activation and disease progression. On the contrary, a recent study conducted in 2021 found an over activity in the mTORC1 pathway in sarcoid granulomas but without any correlation with progression or severity [88]. Of interest, Baker and colleagues found a reduced incidence of sarcoidosis in solid organ transplanted patients when treated with mTOR inhibitors if compared to patients treated with calcineurin inhibitors, suggesting the potential role of mTOR in sarcoidosis pathway (https://doi.org/10.1016/j.semarthrit.2022.152102, accessed on 20 June 2023). Other studies are needed to identify the role of this pathway in the development of pulmonary fibrosis.

### 8.4. Wnt Signaling

The signaling wingless/integrated (Wnt) was discovered for the first time in the carcinogenesis process [89] and its role was extensively studied in other fibrotic diseases like IPF. More specifically, the Wnt-β catenin pathway was found in fibrotic foci of IPF patients [90,91]. In pulmonary sarcoidosis, the Wnt signaling was recently investigated in forty-eight patients, three of which with APS, and compared with eighteen healthy patients [92]. Results demonstrated an altered expression of Wnt5A, Wnt7A and Wnt7B in BAL-fluid cells and an increased β-catenin signaling pathway in sarcoid patients compared to healthy subjects. In another study, authors found a strong correlation between Wnt-β catenin signaling and the epithelial-mesenchymal transition, a mechanism that maintains fibrosis and induces progression [58]. Further studies are needed to clarify the role of Wnt signaling in patients with APS.

## 9. Histological Findings in Fibrotic Sarcoidosis

In advanced pulmonary sarcoidosis, the histological abnormalities maintain the typical distribution along lymphatic routes of pleura, interlobular septa, and bronchovascular bundles, but the typical sarcoid non caseating granuloma loses its chronic lymphocytic infiltrate proceeding towards the fibrotic pathway [93]. Indeed, the granuloma tends to become fibrotic and enlarged, and it is substituted by hyalinized nodes composed by eosinophilic collagen. An important aspect of the pathophysiologic features of fibrotic sarcoidosis is a consistent involvement of small airways and the terminal bronchioles [94]. In APS the chronic lymphocytic inflammation causes bronchial dilatation and small airways distortion leading to mild to severe bronchiectasis [84]. Of interest histologic evaluation from 9 lung explants with end-stage sarcoid lung disease showed dense acellular collagen, granulomas in a lymphatic distribution (along bronchi, the lobular septa, and the pleura), small clusters of macrophages or giant cells nested in fibrotic lesions. Finally, in the end-stage phase of the disease, the histological usual interstitial pneumonia (UIP) pattern could be detected, but differently from patients with Idiopathic Pulmonary Fibrosis, the UIP in APS is predominantly central and with prominent bronchiectasis [26,27].

## 10. Comparison with Other Interstitial Lung Diseases

ILDs and APS are different clinical diseases sharing a progressive deterioration in lung function, physical performance, and quality of life. The diagnosis of sarcoidosis requires compatible clinical and radiological features together with the evidence of non-necrotizing granulomatous inflammation at disease sites. The diagnosis of IPF, the most frequent among the ILD of unknown origin, is reachable after the exclusion of known factors for interstitial lung diseases, and with the presence of radiological and/or radiological pattern of UIP. In the radiological section, we described that a radiological UIP pattern on HRCT similar to IPF may be detected in APS. Based on radiological criteria, UIP pattern in IPF is typically characterized by peripheral and basal predominant reticulations associated to subpleural traction bronchiectasis and honeycombing, whereas in APS fibrotic abnormalities, as well as honeycombing, typically present a peri-bronchovascular and upper lobe predominant distribution. Some patients may present concomitant clinical and radiological features of IPF and sarcoidosis. Combined Sarcoidosis and IPF (CSIPF) may represent a distinct phenotype of these two entities, with different genetic predisposition a different course, or may simply be the coexisting of these two entities in the same patient [95]. Recently a cases series of nine patients with CSIPF were reported and showed a fast functional deterioration during the follow-up period suggesting a worse prognosis similar to patients with IPF [96].

## 11. Treatment

Fibrotic sarcoidosis remains a dangerous and severe disease. The main goals of new therapies are the restoration of the normal M1/Th1–M2/Th2 axis and the prevention of chronic lung infections. As reported in the most recent guidelines, oral glucocorticoids represent the first line treatment in patients with every form of sarcoidosis but evidence that they can prevent progression of pulmonary fibrosis is still scarce [97]. Moreover this treatment is unlikely to benefit patients with stage IV disease due to advanced pulmonary fibrosis [98] and data on other immunosuppressive options, such as methotrexate or infliximab are very limited [99].

In fibrotic sarcoidosis, physicians have to actively look for halting the functional decline and disease progression. Antifibrotic therapy (nintedanib and pirfenidone) has been widely used in IPF and could represent a novel strategy to reduce disease progression also in fibrotic pulmonary sarcoidosis. Recently, the role of nintedanib and pirfenidone has been evaluated in progressive ILD, including fibrotic sarcoidosis [100,101]. In the INBUILD trial [100], progressive ILD patients who received nintedanib, presented a lower functional progression than those who received placebo (FVC ml predicted decline of 80.8 mL vs. 187.8 mL) [100]. Furthermore, the effect of pirfenidone will be explored specifically in stage IV sarcoidosis with more than twenty percent of fibrosis on HRCT scan (ClinicalTrials.gov Identifier: NCT03260556, accessed on 20 June 2023). Lung transplantation could be a surgical option for patients with end-stage fibrotic pulmonary sarcoidosis. Interestingly, among patients with sarcoidosis listed for lung transplant have been reported unexpected sudden death and other causes of mortality [102,103]. 

Lung transplantation is usually a last resort for patients who have a very severe lung damage and reached respiratory impairment due to pharmacological treatment failure. 

Currently, guideline consideration of lung transplant for APS follows those for ILD in general, as there are no specific guidelines for sarcoidosis and includes functional progression, supplemental oxygen requirement, poor lung function and a failure to improve after a trial of medical therapy [104]. This surgical option is reserved for treating patients with very severe organ damage resulting from sarcoidosis. An organ transplant involves removing a damaged organ, such as a lung, and replacing it with a healthy organ from a deceased donor. Organ transplants can significantly improve quality of life or extend life expectancy with a post-transplant survival similar to that in patients with other indications being older age and extensive preoperative lung fibrosis the main factors associated with worse survival [105], but involve many risks, including the risk of infection and even death if the body rejects the transplanted organ.

## 12. Conclusions

The pathogenetic mechanism that leads to pulmonary fibrosis in sarcoidosis is still mysterious and unsolved. Studying different phenotypes and genotypes should be pivotal for increasing our knowledge and for the development of new clinical approaches and therapies. Moreover, further studies should be focused identifying those patients at increased risk of pulmonary fibrosis and disease progression.

## Figures and Tables

**Table 1 ijms-24-10767-t001:** Selected genetic, cells type and signaling associated with fibrotic sarcoidosis as documented in the text.

Features	Role/Function	Type of Studies/Methods	References
**Genes**			
*GREM1 rs1919364 polymorphism*	Patients homozygous for the C allele present a greater risk for fibrosis	Haplotype comparison in human sarcoidosis patients	[35]
*CARD15 rs2066844 polymorphism*	Carriers of the T allele present a greater risk for stage IV CXR	Haplotype comparison in human sarcoidosis patients	[36]
*CCR5 HCC haplotype*	Carriers of both HCC haplotype and *CARD15 rs2066844 T* haplotype always presented stage IV CXR	Haplotype comparison in human sarcoidosis patients	[36]
*PTGS2 rs20417 polymorphism*	Carriers of the C allele present a greater risk for sarcoidosis and a poorer prognosis	Haplotype comparison in human and within sarcoidosis patients	[37]
*TGF-β3 rs3917165 polymorphism*	Carriers of the A allele present a greater risk for fibrosis	Haplotype comparison in human sarcoidosis patients	[38]
*TGF-β3 rs3917200 polymorphism*	Carriers of the C allele present a greater risk for fibrosis	Haplotype comparison in human sarcoidosis patients	[38]
*MUC5B rs35705950 polymorphism*	No association to fibrosis	Haplotype comparison in human sarcoidosis patients	[39]
**Cells**			
*T cells*	T cells are polarized to a Th1/Th17 phenotype, leading to granuloma formation.	Evaluation of Th 17.1 and Th 17 lymphocytes in BALF of sarcoidosis patients compared to control.	[48]
		Evaluation of Th 17.1 in lung mediastinal lymph nodes of sarcoidosis patients.	[49]
		T cells evaluation in BALF and blood of sarcoidosis patients.	[50]
*Macrophages*	evidence suggests the transition from a M1 to a dominant M2 phenotype in more advanced stages of sarcoidosis	Evaluation of M2 macrophages in tissues specimens of sarcoidosis patients compared with tuberculosis patients. In vitro comparison of PBMCs from sarcoidosis patients and controls	[51,52]
**Signaling**			
TGF-β/SMAD signaling	Key role during inflammatory mitigation and wound healing. Three forms of TGF-β: TGF-β1: strongly connected with fibrosis; TGF-β2: implicated in post inflammatory wound healing; TGF-β3: role still unsolved.	In vitro evaluation on human AEC Transcription evaluation in cells from pulmonary fibrosis patients and controls Gene expression evaluation in BALF cells and blood lymphocytes from sarcoidosis patients	[53,54,55]
JAK-STAT signaling	Regulates the release of INF-γ. The pathway is over expressed in patients with sarcoidosis.	microRNA expression in PBMCs of controls and sarcoidosis patients	[56]
mTOR signaling	mTOR signaling promotes collagen synthesis in the lung fibroblasts through aerobic glycolysis and represents a powerful autophagy inhibitor. Its role in sarcoidosis is still unclear.	In vitro evaluation on human lung fibroblasts	[57]
Wnt signaling	This pathway seems to be associated with fibrosis and progression.	Gene expression in lung sample from fibrotic patients and controls, in vitro evaluation of gene downregulation in human AECs	[58]

## Data Availability

No new data were created.

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
