# Peer review of "Molecular Mechanism in the Development of Pulmonary Fibrosis in Patients with Sarcoidosis"

_ijms, 2023, doi:10.3390/ijms241310767_

Round 1

Reviewer 1 Report

I would congratulate for this very good paper, documenting the clinical-prognostic characteristics and molecular pathways which are believed to be associated with the development of Advanced Pulmonary Sarcoidosis.. Discussion section: In sarcoidosis population, specially considering pulmomary- cardiac relationships,  the authors should more discuss  the impact of cardiac sarcoidosis. Cardiac involvement in sarcoidosis often manifests in a variety of ways that including congestive heart failure with pulmonary impact. Additionally, sarcoidosis can mimic other cardiac conditions, such as HCM. In a cardiac sarcoidosis population Matsumori et al reported 7% of patients showing echocardiographic abnormalities (DOI: 10.1253/jcj.64.679) While echocardiography has a low sensitivity in the detection of sarcoidosis, authors should more discuss the importance of cardiac magnetic resonance in HCM differential diagnosis allowing detection of edema and abnormal scar patterns (DOI: 10.1016/j.ijcard.2021.10.013). Not by chance, while definitive diagnosis of cardiac sarcoidosis can be made with endomyocardial biopsy, myocardial involvement is often patchy in nature, and it may be reasonable to identify suitable lesions by diagnostic imaging prior to biopsy to increase sensitivity. Finally authors should more focus on cardiac sarcoidosis symptoms (DOI: 10.3389/fmed.2023.999066) also considering potential differential diagnosis with HCM ( DOI: 10.1016/j.ijcard.2022.03.028). With no specific symptoms, cardiac sarcoidosis may be difficult to suspect in a patient with no previous extra-cardiac sarcoidosis diagnosis. Please add all the 4 suggested references

Author Response

We thank the reviewer for the nice comment and useful advice. She/he is correct, cardiac sarcoidosis has a relevance in prognosis of the affected patients, so we have e added in the ‘introduction’ section the reference you suggested.However, since the cardiac involvement in sarcoidosis is not the core of our review, we’ve chosen not to further describe symptoms and radiological evaluation of this sarcoidosis manifestation.

Reviewer 2 Report

The aim of this manuscript is to summarize the clinical prognostic features and molecular pathways, probably associated with the onset and progression of sarcoidosis.

This manuscript shows rich content, providing a deep insight for some works: the study is within the journal’s scope, and I found it to be well-written, providing sufficient information. Even if the manuscript provides an organic overview, with a densely organized structure and based on well-synthetized evidence, there are some suggestions necessary to make the article complete and fully readable. For these reasons, the manuscript requires major changes.

Please find below an enumerated list of comments on my review of the manuscript:

INTRODUCTION:

The authors should provide a list of the abbreviations, mentioned in the manuscript, to improve the clarity and quality of the manuscript.

LINE 40: During COVID-19 pandemic, sarcoidosis patients have been considered an at-risk population, due to the impaired lung function and fibrosis in multiple organ system (see, for reference: Torge, D.; Bernardi, S.; Arcangeli, M.; Bianchi, S. Histopathological Features of SARS-CoV-2 in Extrapulmonary Organ Infection: A Systematic Review of Literature. Pathogens 202211, 867. https://doi.org/10.3390/pathogens11080867). By virtue of the recent COVID-19 pandemic, the authors should highlight in this introductive section the association between sarcoidosis and multiple organ system damage, observed in SARS-CoV-2 infection.

LINE 324: mTOR may protect against the development of sarcoidosis, as suggested by the lower incidence of this disease in solid organ transplant patients, receiving mTOR inhibitors (see, for reference: Baker, M. C., Vágó, E., Liu, Y., Lu, R., Tamang, S., Horváth-Puhó, E., & Sørensen, H. T. (2022, December). Sarcoidosis incidence after mTOR inhibitor treatment. In Seminars in Arthritis and Rheumatism (Vol. 57, p. 152102). WB Saunders).

The main topic is interesting, and certainly of great clinical impact. As regards the originality and strengths of this manuscript, this is a significant contribute to the ongoing research on this topic, as it extends the research field on the clinical prognostic features and molecular pathways, probably associated with APS onset and progression. Overall, the contents are rich, and the authors also give their deep insight for some works.

Furthermore, the manuscript relies on a multitude of scientific evidence, to derive its conclusions. The results are reliable and adequately discussed.

The conclusion of this manuscript is perfectly in line with the main purpose of the paper: the authors have designed and conducted the study properly. As regards the conclusions, they are well written and present an adequate balance between the description of previous findings and the results presented by the authors.

In conclusion, this manuscript is densely presented and well organized, based on well-synthetized evidence. The authors were lucid in their style of writing, making it easy to read and understand the message, portrayed in the manuscript. Besides, the methodology design was appropriately implemented within the study. However, many of the topics are very concisely covered. This manuscript provided a comprehensive analysis of current knowledge in this field. Moreover, this research has futuristic importance and could be potential for future research. However, I have major comments for this manuscript, for improvement before acceptance for publication. The article is accurate and provides relevant information on the topic and I have some major points to make, that may help to improve the quality of the current manuscript and maximize its scientific impact. I would accept this manuscript if the comments are addressed properly.

Author Response

We kindly appreciate this comment that helps to clarify our work. We created the list of abbreviations and we added it after the abstract section (lines 25 to 36)

2 - LINE 40: During COVID-19 pandemic, sarcoidosis patients have been considered an at-risk population, due to the impaired lung function and fibrosis in multiple organ system (see, for reference: Torge, D.; Bernardi, S.; Arcangeli, M.; Bianchi, S. Histopathological Features of SARS-CoV-2 in Extrapulmonary Organ Infection: A Systematic Review of Literature. Pathogens 2022, 11, 867. https://doi.org/10.3390/pathogens11080867). By virtue of the recent COVID-19 pandemic, the authors should highlight in this introductive section the association between sarcoidosis and multiple organ system damage, observed in SARS-CoV-2 infection.

Thanks for the useful comment regarding sarcoidosis patients during COVID-19 pandemic, we added a sentence regarding the issue of multisystemic involvement of both sarcoidosis and SARS-CoV-2 infection in the introduction, (Lines 40 to 42).

LINE 324: mTOR may protect against the development of sarcoidosis, as suggested by the lower incidence of this disease in solid organ transplant patients, receiving mTOR inhibitors (see, for reference: Baker, M. C., Vágó, E., Liu, Y., Lu, R., Tamang, S., Horváth-Puhó, E., & Sørensen, H. T. (2022, December). Sarcoidosis incidence after mTOR inhibitor treatment. In Seminars in Arthritis and Rheumatism (Vol. 57, p. 152102). WB Saunders).

We profoundly thank the reviewer for this insightful suggestion that adds value to our dissertation. We've introduced the paper in the mTOR pathway section (line 277 to 280).

Round 2

Reviewer 1 Report

Author's did not respond to this reviewer's suggestions, including the fundamental relationship with cardiac sarcoidosis ( sarcoidosis-HCM and structural heart disease) nor the 4 references references. 

Author Response

We thank the reviewer for the congratulation regarding the review we have submitted to the journal which is focused on clinical-prognostic characteristics and molecular pathways associated with the development of Advanced Pulmonary Sarcoidosis. Following your suggestion we included  in the introduction 2 references  (out of 4 references) in particular we mentioned the ones more appropriate to the aim of the review.

Reviewer 2 Report

The authors have significantly improved the manuscript. I accept it for the publication. 

Author Response

We thank the reviewer for the congratulation regarding the review we have submitted to the journal.